# Involved-Site Radiation Therapy Enables Effective Disease Control in Parenchymal Low-Grade Primary Cerebral Lymphoma

**DOI:** 10.3390/cancers15235564

**Published:** 2023-11-24

**Authors:** Niklas Benedikt Pepper, Michael Oertel, Gabriele Reinartz, Khaled Elsayad, Dominik Alexander Hering, Fatih Yalcin, Moritz Wildgruber, Walter Stummer, Georg Lenz, Wolfram Klapper, Hans Theodor Eich

**Affiliations:** 1Department of Radiation Oncology, University Hospital Muenster, 48149 Muenster, Germany; 2Department of Pathology, University Hospital Schleswig-Holstein, 24105 Kiel, Germany; 3Department for Radiology, University Hospital Munich, LMU Munich, 81377 Munich, Germany; 4Department of Neurosurgery, University Hospital Muenster, 48149 Muenster, Germany; 5Department of Hematology, Oncology and Pneumology, University Hospital Muenster, 48149 Muenster, Germany

**Keywords:** lymphoma, whole-brain irradiation, cranial radiotherapy, PCNSL, marginal-zone lymphoma, follicular lymphoma, ISRT, involved-site radiotherapy

## Abstract

**Simple Summary:**

Malignant lymphomata originating from the central nervous system are rare entities and usually are characterized by rapid growth, requiring intensified treatment. In contrast, low-grade lymphoma variants may be treated using radiotherapy alone. As the majority of cases arise from the dura, only very few cases of parenchymal low-grade lymphoma have been reported in the literature. We present a retrospective analysis of two patients treated with involved-site radiotherapy at our institution and discuss the available literature. Overall, these patients have a good prognosis and may be treated using primary radiation therapy.

**Abstract:**

Background: Primary lymphoma of the central nervous system (PCNSL) encompasses a variety of lymphoma subtypes, with the majority being diffuse large B-cell lymphomas, which require aggressive systemic treatment. In contrast, low-grade lymphomas are reported infrequently and are mostly limited to dural manifestations. Very rarely, parenchymal low-grade PCNSL is diagnosed, and the cases documented in the literature show a wide variety of treatment approaches. Methods: We screened all cases of PCNSL treated at our department (a tertiary hematooncology and neurooncology center) in the last 15 years and conducted a comprehensive literature research in the PubMed database. Results: Overall, two cases of low-grade primary parenchymal PCNSL treated with irradiation were identified. The dose prescriptions ranged from 30.6 to 36 Gy for the involved site, with sparing of the hippocampal structures. Both patients had an excellent response to the treatment with a mean follow-up of 20 months. No clinical or radiological signs of treatment toxicity were detected. Conclusions: Our analysis corroborates the results from the literature and demonstrates that parenchymal low-grade PCNSL shows a good response to localized radiation treatment, enabling a favorable outcome while avoiding long-term treatment toxicity.

## 1. Introduction

Primary lymphoma of the central nervous system (PCNSL) is a rare diagnosis with an overall poor patient outcome. Accounting for approximately 3–4% of all cerebral malignancies [1,2,3], it can affect all components of the CNS, including the meninges, eyes, parenchyma, spinal cord, the choroid plexus and spinal fluid. Symptoms may vary depending on the site of the disease, with 70% involving focal neurological deficits and 43% non-specific neurocognitive symptoms [3]. Confirmation of diagnosis is obtained using stereotactic biopsy (if feasible) with diffuse large B-cell lymphoma (DLBCL) being the predominant histopathological subtype [1,2]. Additional staging also includes ophthalmological assessment, spinal cord MRI, bone marrow biopsy and whole body FDG-PET/CT to rule out systemic disease or primary extracerebral sites [3].

High-dose methotrexate (HD-MTX) is the chemotherapeutic regime of choice, but with overall low case numbers due to the rarity of PCNSL, the optimal treatment approach is yet to be determined [2]. Even with improvements in treatment response and aggressive treatment regimes, long-term disease remission is only achieved in under 50% of patients [3,4]. Historically, radiotherapy (RT) played a pivotal role in the treatment of PCNSL, due to the high radiosensitivity of the disease. Whole-brain irradiation (WBI) has been the standard of care and still plays a role in consolidation after initial chemotherapy. Recently, the value of RT has been questioned in light of delayed neurocognitive impairment as a result of WBI [4,5,6,7]. At the moment, clinical trials aim to reduce treatment toxicity by deescalating the RT dose/or avoiding RT entirely [8,9].

Low-grade lymphoma also occurs within PCNSL (LG-PCNSL), but only represents a small fraction of cases. A retrospective analysis of PCNSL cases by Karschnia et al. of patients treated in the divisions of neuro-oncology at Massachusetts General Hospital and the Yale School of Medicine found only 6.3% of primary cerebral lymphoma to be dural lymphoma, with marginal-zone lymphoma being the most common subtype amongst them [10]. A German meta-analysis of three large trials involving patients with PCNSL identified 3% of patients with low-grade histological subtypes, mainly marginal-zone lymphoma, almost exclusively in the dura [11]. A retrospective French analysis by Desjardins et al. of patients treated at Pitié-Salpêtrière Hospital also detected a low rate (1.6%; 11 patients) of low-grade lymphoma in a group of 662 patients with PCNSL. Again, the majority (*n* = 8) were dura-based, with the remaining three cases being located in the parenchyma [12]. Other publications on LG-PCNSL consist of case reports with varying depths of detail (histopathologic description, imaging, treatment) [13,14,15,16,17,18]. Low case numbers and complex differential diagnoses such as CNS involvement in systemic lymphoma, including rare subtypes like Bing–Neel syndrome (BNS, a rare neurologic complication of lymphoplasmocytic lymphoma (LPL)), complicate the process of diagnosis [19].

With indolent lymphoma being extremely rare, no standard of care has been established. Since low-grade lymphoma such as follicular lymphoma and marginal-zone lymphoma are known to respond well to local RT [20,21], aggressive chemotherapy might not be necessary to achieve favorable outcomes. We present an analysis of two patients with parenchymal low-grade lymphoma treated using monomodal radiation therapy at our department and analyze the data in light of a comprehensive review of the literature. Thereby, insights on the optimal radiation–oncological treatment strategy for LG-PCNSL are provided.

## 2. Materials and Methods

### 2.1. Patient Selection

We retrospectively reviewed all cases of patients with PCNSL treated at the radiation oncology department of our university hospital between 2007 and 2022. Records were screened for accessibility of treatment planning, diagnostic and follow-up imaging, as well as histopathological samples with elaborate diagnosis and follow-up records.

### 2.2. Radiotherapy Planning and Administration

The radiotherapy for all patients was realized using the Varian Eclipse^TM^ treatment planning system (TPS) and the TrueBeam^®^ (Varian Medical Systems, Palo Alto, CA, USA) linear accelerator (LINAC) or tomotherapy TPS TomoHelical with TomoTherapy^®^ (Accuray Inc., Sunnyvale, CA, USA). The RT plans were executed using 6 MV photons in intensity-modulated radiotherapy/volumetric modulated arc therapy (IMRT/VMAT) or helical therapy, as a form of volumetric modulated arc therapy. For IMRT/VMAT planning, the clinical target volumes were defined as standardized by ICRU-50/ICRU-83, with the GTV enclosing the primary disease site, marked by radiographical changes in contrast in enhanced T1-weighted and T2-weighted MRI. The CTV marked sites of supposed subclinical infiltration and the PTV created an additional margin to compensate for possible incongruences in patient positioning (3–5 mm). Immobilization was enabled by the use of thermoplastic masks and image guidance was performed using cone-beam CT/megavoltage CT on a daily basis.

### 2.3. Follow-Up Data

All accessible patient data of clinical follow-up visits regarding the treatment response and toxicities were collected from the clinical files and the hospital’s information system (ORBIS, Dedalus Healthcare, Bonn, Germany), which provided clinical and radiological re-assessment, toxicity documentation, doctors’ letters and imaging. The follow-up schedule included oncological re-assessment, with the evaluation of local and/or systemic relapse via cranial MRI, blood work and clinical and sonographical examination of the lymph nodes, and radio-oncological toxicity assessments on a quarter-annual basis, which were extended to bi-annual visits after 2 years if no conspicuous findings arose. There was a particular focus on any signs of neurological toxicity after radiotherapy, such as neurocognitive symptoms (impaired memory, difficulties concentrating, prolonged fatigue, decelerated cognition), the occurrence of or alteration in motor impairment and sensory adverse effects (vertigo, deterioration of hearing or vision). All follow-up MRI imaging was reviewed and the lymphoma remission status reassessed critically. A secondary retrospective re-assessed by an expert radiologist regarding the initial imaging presentation of the disease and response to treatment was conducted. MR imaging was routinely performed at 3 T applying T1-weighted 3D sequences.

Reports were screened for radiological changes as a possible manifestation of treatment toxicity, such as microangiopathy, leukoencephalopathy or cerebral atrophy.

### 2.4. Histopathological Verification of Diagnosis

For all patients, initial diagnosis was performed based on tissue biopsy. All cases were initially processed using local neuropathology and reassessed for confirmation of the diagnosis by an expert in lymphoma pathology. A diagnosis of low-grade PCNSL was made based on the morphological tumor cell appearance, immunohistochemical profile and, if necessary, molecular pathological verification. The immunohistological staining included CD45, CD3, CD20, CD5, CD10, CD79a, CD23, Cyclin D1, CD30, MUM1, BCL2, BCL2, Ki-67 and lambda/kappa light chain in situ hybridization (-ISH). For differential diagnostics to LPL, we additionally directly sequenced the MYD88 gene (pyrosequencing) and CXCR4 gene (Sanger sequencing). The sequencing results showed no mutations at codon position 265 of the MYD88 gene codon or 291 to 352 of the CXCR4 gene codon.

### 2.5. Literature Review

A comprehensive literature search was conducted in the NCBI PubMed database, using the search term “((marginal zone lymphoma [All] OR follicular lymphoma [All] OR indolent lymphoma [All]) AND brain [MeSH Terms]) OR ((PCNSL [All] OR intracranial [All] OR primary cerebral [All]) AND (marginal [All] OR indolent [All] OR follicular [All]) AND lymphoma [MeSH Terms])”. All entries were screened for the availability of the abstract in English, covering a timespan from 1967 to 2023. All accessible abstracts were then reviewed on whether they were fitting for the subject at hand and excluded if the article did not feature PCNSL (but, for example, cerebral relapse or manifestation of systemic lymphoma), low-grade disease (but DLBCL) or parenchymal manifestation of lymphoma. Articles not providing data concerning the treatment or outcome of patients or purely reviewing the literature without any additional patient data were also excluded. If in doubt regarding one or more of the inclusion criteria, a full text evaluation was performed. The remaining articles were all assessed regarding the individual patient data at least for the applied treatment regimens and patient outcome, but also the radio-morphological appearance, immunohistochemistry and recorded treatment toxicity. The last date of the literature review was 15 April 2023.

## 3. Results

### 3.1. Patient Demographics and Data

A cohort of 88 patients with PCNSL was identified and reviewed for the fit of the inclusion criteria. Two cases met the criteria of low-grade PCNSL located in the parenchyma, treated using radiotherapy. Both cases were diagnosed as MZBCL, while another case of suspected FL was later classified as DLBCL and removed from the pooled analysis. One patient was initially diagnosed with bone marrow involvement and treated with four cycles of rituximab (RTX). Sequential restaging showed a partial response with no further detection of lymphoma cells in the bone marrow, but progressive disease of the parenchymal manifestation of the left parietal lobe (see Figure 1). The demographics, diagnostic and treatment characteristics and response details are summarized in Table 1.

### 3.2. Radiotherapy

After CT- and MRI-based treatment planning, both patients received daily RT of 1.8 Gy per fraction using six MV photons via local conformal irradiation of the tumor site (involved-site radiotherapy (ISRT)). A female patient with MZBCL of the left frontal lobe received 36.0 Gy in 20 fractions, using VMAT with two coplanar arcs. In this case, the CTV was defined as a margin of 10 mm around the contrast-enhancing areas, also incorporating T2-weighted tissue abnormalities with an additional 5 mm margin for the PTV.

For the other patient, the treatment volume sizes were further reduced to a CTV with a 5 mm margin around the contrast-enhancing areas, again incorporating T2-weighted signs of local tissue alterations, and an additional 3 mm to create the PTV. RT was realized using helical therapy via tomotherapy in 17 fractions of 1.8 Gy to a prescription dose of 30.6 Gy. An example of the target volume definition and dose distributions of the resulting plan is given in Figure 2.

### 3.3. Follow-Up Data

Each patient was subjected to regular oncological as well as radio-oncological follow-up examinations assessing the treatment response and toxicity. Imaging via contrast-enhanced cranial MRI scans was applied accordingly. Overall, 13 quarter-annual follow-up visits and 8 MRI scans were evaluated (8 follow-ups and 5 MRIs for patient 1, 5 follow-ups and 3 MRIs for patient 2). The MRI work-up included T1- and T2-weighted sequences in the coronary, sagittal and transversal reconstruction, as well as diffusion-weighted sequences (TRACE and ADC). The initial treatment was tolerated well in all patients with no side effects surpassing moderate fatigue and local alopecia. Both patients showed no signs of neurocognitive decline, corresponding to overall low doses to the hippocampal region under 8 Gy D_mean_. One patient received a 13 Gy mean dose to the ipsilateral hippocampus and a 2.9 Gy mean dose to the contralateral hippocampus and did not show any impairment in the follow-up examinations (14 months after treatment). MRI scans show stable disease after local treatment. In the case of the patient treated with 30.6 Gy ISRT, a reduction in the contrast agent uptake in the first follow-up MRI was rated as a partial response. This patient also revealed an increase in KPS, based on a severe reduction in the neurological symptoms (mainly headache) that were prevalent before treatment. No patient showed local or distant relapse or progression during follow-up.

### 3.4. Histopathological Diagnosis and Immunohistochemistry

Both cases were diagnosed as MZBCL after careful exclusion of other diagnoses such as lymphoplasmacytic lymphoma, another low-grade B cell lymphoma characterized by small lymphocytes. In both presented cases, the lymphocytic infiltrate was very subtle and was found in fairly sharply defined clasp-like arrangements around the vessels. Cytologically, the infiltrate was lymphocytic/small cellular (Figure 3A,B). The immunohistology showed the lymphoma cells as CD20-positive (Figure 3C,D) with a low Proliferation Index (Figure 3E). The immunohistology for the presented patient displayed negative staining for Cyclin D1, CD23 and BCL6 (Figure 3F,I,J), as well as a kappa light chain restriction (Figure 3G,H).

### 3.5. Literature Review

The systematic review of the NCBI PubMed database identified 143 results with the designed search string. All records were screened; one article was excluded because neither the abstract nor the full text was available in English. All other abstracts were assessed, if necessary, with further investigation of the full text. After the review, a total of 133 articles were excluded. These articles did not feature cases of PCNSL (57 articles), did not include low-grade lymphoma (24 articles), did not feature parenchymal disease location (42 articles with 39 articles describing dural manifestations of low-grade lymphoma and 3 articles describing lymphoma of the choroid plexus), did not provide original data (four reviews without additional patient data) or did not describe the treatment procedures or patient outcome (five articles only featuring diagnostic findings). Figure 4 shows the workflow of the review process as a PRISMA flow diagram and Table 2 lists the significant results for comparison with the cohort at hand.

The remaining 10 articles were reviewed for full text eligibility, which applied to all of them, and all information regarding demographics, histopathology, treatment and sequential response in follow-up was gathered for discussion in context of the data given in this work. Of note, the largest cohort with detailed data regarding diagnostic workup and therapy is presented by Jahnke et al. [11], describing 10 cases of parenchymal lymphoma. Table 2 shows a selection of the data, tailored to fit the immunohistological marker profile of the patients analyzed in this article, selecting MZBCL and FL cases [22]. The results depict a diverse conglomerate of small cohort series (similar to this report) treated with different approaches using RT and/or chemotherapy and covering a latitude of outcomes and toxicities. The corresponding publications, as well as their resulting implications in the context of this analysis, will be discussed in the following segment to deduce a conclusive interpretation of the existing data.

**Table 2 cancers-15-05564-t002:** Database. Literature data: results of a comprehensive literature research of parenchymal low-grade lymphoma of the CNS. For studies with multiple patients, only cases of parenchymal low-grade BCL were included.

Authors	Year	Patients Overall (Parenchymal)	Selected Patients	Histological Subtype	Sex	Age	RT Concept (Dose/Fractions)	System Therapy	Reported Neurological Toxicity	Response (Months)
Jahnke et al. [11]	2004	10 (10)	1 2 3	BCL NOS BCL NOS LPL	M F M	60 61 58	- - -	HD-MTX HD-MTX HD-MTX	Neuropsychological Neuropsychological Epilepsy	CR (44.5+) SD (22.5+) CR (14.5+)
			4 5 6 7	FL BCL NOS LPL BCL NOS	M F M M	60 60 19 45	WBI (N/A) - - WBI (N/A)	BMPD + iMTX + CHOP HD-MTX HD-MTX HD-MTX	Focal motor + Neuropsychological - - -	CR (54+) CR (58+) SD (33.5+) CR (10+)
Tu et al. [17]	2005	15 (1)	1	MZBCL	M	66	WBI (N/A)	-	N/A	CR (13+)
Park et al. [13]	2008	1	1	MZBCL	M	18	ISRT (30.6/17)	-	-	CR (22+)
Lim et al. [14]	2011	15	1	MZBCL	M	57	WBI	MTX	N/A	CR (39)
		(1)					(30.6/17)			
Papanicolau et al. [23]	2011	1	1	MZBCL	M	70	-	TMZ + RTX	-	SD (N/A)
Aquil et al. [18]	2012	1	1	MZBCL	M	48	WBI (24/12) + 6 Gy Boost	none	-	PR (15+)
Schiefer et al. [22]	2012	1	1	MZBCL	F	39	-	HD-MTX + Ara-C	-	SD (24+)
Ueba et al. [15]	2013	1	1	MZBCL	M	53	-	MTX + Ara-C	N/A	CR (N/A)
Epaliyanage et al. [16]	2014	1	1	MZBCL	F	58	IRST N/A	-	N/A	SD (24)
Desjardins et al. [12]	2022	11 (3)	1 2 3	MZBCL MZBCL MZBCL	F M F	57 58 53	- ISRT (40/20) -	R-F R-F HD-MTX	- - -	PR (47.9) SD (55.2+) SD (110+)

Abbreviations: BCL NOS: B cell lymphoma not otherwise specified, LPL: lymphoplasmocytic lymphoma, CR: complete remission, FL: follicular lymphoma, ISRT: involved-site radiotherapy, MZBCL: marginal-zone B cell lymphoma, HD-MTX: high-dose methotrexate, N/A: data not available, PR: partial remission, R-F: Rituximab-fludarabine, RTX: rituximab, SD: stable disease, TMZ: temozolomide, WBI: whole-brain irradiation.

## 4. Discussion

The presented analysis is one of the few structured reports on the role of radiotherapy for primary low-grade CNS lymphoma. It reveals that:(1)ISRT is a feasible and effective treatment option for this rare entity.(2)Individualized local fields seems sufficient to control the disease with no need for whole-brain irradiation.(3)Local RT did not result in neurocognitive decline during the period of observation.

In general, very few cases of this rare entity have been reported, with dural manifestations representing the majority of these cases, oftentimes initially mistaken for meningioma and treated using resection [14,24,25], while parenchymal manifestations are even less frequent [11,12,14]. Our analysis has demonstrated that the reported cases were treated with different approaches to therapy. Additionally, one patient was initially included in this cohort based on an initial diagnosis of cerebral FL, but later on was removed due to referential pathologic classification as DLBCL: this patient received RT as WBI with 45.0 Gy (25 fractions) with a sequential dose escalation (“boost”) to the primary tumor site of 16.0 Gy in eight fractions (2.0 Gy per fraction). The CTV for the boost was defined with a 20 mm margin around the GTV, defined as the area with contrast-enhanced alterations in T1-weighted MRI, with an additional 5 mm for the PTV. Treatment was realized as 3D-CRT with lateral opposing fields for WBI and a four-field box technique for the boost, with opposing lateral and anterior–posterior radiation beams using the LINAC Primus. He discontinued radio-oncological reassessment after 39 months and solely engaged in regular oncological follow-up, showing a complete response after radiotherapy without any relapse for over 40 months. Regarding toxicity, this patient, in contrast to both patients with localized RT, showed radiographical and clinical signs of treatment toxicity. After the initial improvement of neurologic symptoms (insecure walking), he reported signs of neurocognitive deterioration in the form of impaired short-term memory and trouble concentrating, leading to a decline in KPS with a greater demand for assistance in everyday settings. These symptoms were clinically apparent at the first 24 months after RT. MRI scans showed bi-hemispheric leukoencephalopathy.

PCNSL is usually treated aggressively using systemic chemotherapy, the backbone of which is HD-MTX [2,26]. As seen in one excluded patient, the identification of low-grade variants itself is complex and requires interdisciplinary assessment with the consultation of experts (in pathology and radiology), with otherwise possible over-treatment using aggressive approaches. Desjardins et al. presented a larger cohort of patients with parenchymal low-grade lymphoma treated using chemotherapy only or a combined modality treatment, including three cases of parenchymal MZBCL with an overall good response (mean PFS of 78 months and 10-year OS of 90%) [12].

However, it is known that lymphoma in general, and especially low-grade lymphoma such as MZBCL or FL, is highly responsive to RT alone [20,21,27,28,29]. This turned out to be true for parenchymal LG-PCNSL in the patient cohort presented here and is corroborated by the literature [11,12,13,14,16,23,30]. In accordance, a SEER database analysis of 4375 PCNSL patients concluded that for patients with indolent PCNSL, radiation monotherapy is an appropriate treatment concept, leading to prolonged survival [31]. Moreover, the analysis by Nomani et al. suggests that RT for parenchymal manifestations of marginal-zone lymphoma showed a superior outcome when measured against different regimes of chemotherapy, based on a series of 11 cases of cerebral PCNSL. Unfortunately, no details are given regarding the field design or dose of RT, but stable disease for 61 months after diagnosis was reported [32].

On the matter of treatment response, Corn et al. found a strong correlation between CR as measured in imaging and increased survival for patients with PCNSL [33], but this does not necessarily seem to apply to LG-PCNSL. This was also already suggested by Desjardins et al. [12], raising the question of factual CR despite residual contrast enhancement after therapy. The work of Jahnke et al. with a long observation period has also previously demonstrated that stable disease might be sufficient and not associated with prognosis. This also seems to be valid in other cases of cerebral manifestations of indolent lymphoma like BNS as a possible differential diagnosis of LG-PCNSL.

BNS, a rare neurological complication of LPL (also named Waldenström’s macroglobulinemia (WM)), oftentimes features diffuse neurological symptoms. Due to its rarity, the optimal therapeutic approach is yet to be defined. In a 2017 guideline for the diagnosis and treatment of BNS, proposed by an interdisciplinary task force assembled during the 8th International Workshop on WM, a symptom-oriented approach is recommended [19]. Similar to LG-PCNSL, residual tissue alterations after treatment do not seem to necessitate immediate salvage therapy. Despite reports of a good response to RT [34,35], the authors chose to omit irradiation from first-line treatment considerations based on the apprehended neurocognitive side effects. This concern might be valid for extensive treatment fields, but modern approaches with reduced margins and dose concepts might greatly improve the value of RT in this context as well, and should thus be considered in the future.

Overall, the value of cranial irradiation in lymphoma treatment seems to be underestimated based on poor experience with large treatment fields in the past, which led to the limitation of RT use in PCNSL treatment. The previously commonly applied WBI has been challenged due to its neurocognitive toxicity, especially in the elderly, and modern guidelines aim to postpone the use of extensive cerebral irradiation for as long as possible [2]. While long-term cognitive impairment is a serious concern, especially in patients with an overall good treatment response and prognosis, HD-MTX can also lead to changes in the white matter brain structure and neurocognitive impairment or psychosocial deterioration [36,37]. This is also reflected in the work of Jahnke et al., showing several cases with neuro-psychosocial deterioration after HD-MTX alone [11]. To avoid cognitive impairment, de-escalation concepts such as low-dose WBI were introduced into PCNSL treatment, based on dose–side effect correlation [2,8]. In our analysis, the only patient developing memory impairment and trouble concentrating was treated using WBI (and later excluded based on classification as DLBCL). Additional use of RT after HD-MTX, especially WBI, seems to further increase the risk of neurocognitive sequelae [5,38]. Examples may also be found in a patient in the abovementioned series (Table 2, Jahnke et al., patient 4) and in the work of Dejardins et al., describing a 77-year-old patient with neurological toxicity after RT who underwent two lines of prior (immuno-)chemotherapy (rituximab-based and HD-MTX) [11,12].

Another strategy to reduce the neurological side effects in cranial irradiation may be the de-escalation of the treatment field size. Our work demonstrates the use of limited-size ISRT along with small margins to be efficient and feasible, leading to lasting disease control. ISRT as a form of local treatment, sparing uninvolved brain matter and leading to at least SD, has been demonstrated in several reports analyzed in this work [13,16] as well. In general, a reduction in the radio-oncological target volumes and prescription doses in lymphoma treatment results in better tolerance of treatment without compromising the results in several forms of lymphoma [39,40,41]. Novel and improved methods of imaging, as well as more precise delivery systems and techniques for RT, using conformal image-guided therapy, are the main underlying advancements. Additional sparing of critical structures in cranial irradiation has become a topic of interest in modern-day RT as well. Modern approaches to cranial irradiation with hippocampal sparing have shown advancements in the preservation of cognitive function and are adopted in the treatment of different disease entities [42,43,44]. Their role in local (ISRT) or extended (WBI) cranial irradiation for lymphoma needs to be explored in the future.

A concern of solely local treatment might be systemic relapse as a form of treatment failure. Iwamoto et al. reported three cases with systemic relapse after a median of 6.8 years in a series of eight patients with CNS-based MZBCL after initial CR [30,45]. In such cases, systemic treatment is a valuable asset (for example, using Zanubrutinib). One patient in our cohort also required systemic therapy: after an initial lack of treatment due to prolonged diagnosis, she developed systemic disease with bone marrow involvement, showing an incoherent response with complete systemic remission after HD-MTX, but progressive disease of the primary lymphoma site in the left parietal lobe. This patient, as well as the others, responded well to local RT with partial remission at the primary site and has since displayed stable disease under close monitoring.

In conclusion, we advocate for the use of tailored ISRT with limited margins for the treatment of parenchymal low-grade PCNSL since this approach not only offers long term disease control and seems to be tolerated well (as reflected in our cohort and the literature), but also spares the patient from undergoing high-dose chemotherapy. De-escalation of treatment in this favorable form of PCNSL is needed to avoid long-term toxicity. However, evidence for different regimes is limited due to the overall small cohort sizes. Due to the low recruitment prospects, randomized clinical trials for this entity seem unlikely to happen in the future.

Another limitation is the comparability of the data, hindering generalized evidence due to varying treatment concepts (WBI vs. ISRT with varying margins). These originate from rapidly advancing technologies in modern radiotherapy with improved possibilities of delivering precise treatment (leading to smaller margins), as well as increased data allowing for the de-escalation of radiation doses and treatment field sizes. In the future, the benefit of conformal ISRT should be discussed as an option for cerebral lymphoma manifestations like LG-PCNSL and not be omitted in fear of neurocognitive sequelae. Other entities like LPL might also benefit from this approach.

Ideally, all gathered data regarding this rare group of entities should be pooled in a register study and published to broaden the spectrum of clinical experience to be discussed in future decision-making, as presented in this paper.

## 5. Conclusions

Involved-site radiotherapy is an effective and well-tolerated treatment option for patients with LG-PCNSL, offering local control without the need for high-dose systemic chemotherapy. Radiotherapy in LG-PCNSL should be used upfront in a timely manner. Delaying therapy should be avoided in order to prevent the development of systemic spread.

## Figures and Tables

**Figure 1 cancers-15-05564-f001:**
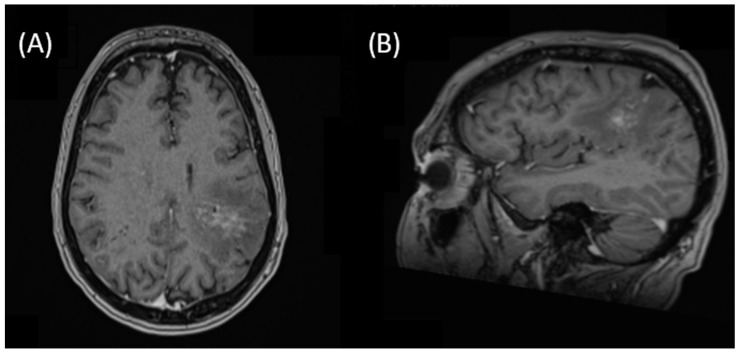
Unusual presentation of parenchymal PCNSL of the left parietal lobe, later confirmed as MZBCL in transverse (**A**) and sagittal (**B**) contrast-enhanced T1-weighted MRI. The pattern is consistent with the radiologic presentation previously described by Epaliyanage et al. (“brain on fire sign”) [16].

**Figure 2 cancers-15-05564-f002:**
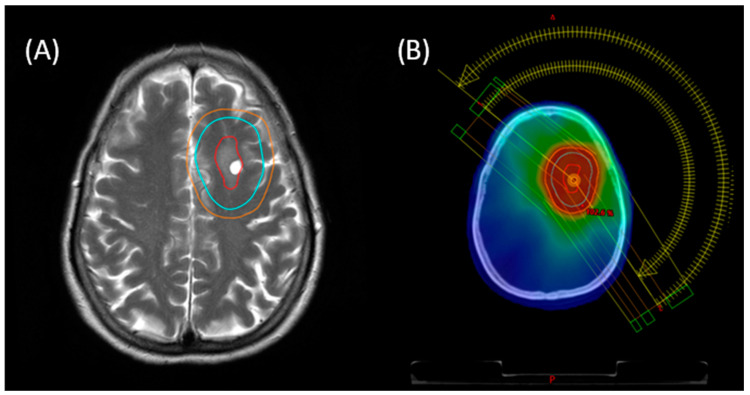
Treatment planning with dose distribution for MZBCL of the left frontal lobe (axial view). (**A**) Planning CT with co-registered planning MRI with GTV (red), CTV (cyan) and PTV (orange). (**B**) Dose distribution and arc setup of the resulting treatment plan of 36.0 Gy ISRT in 20 fractions via VMAT.

**Figure 3 cancers-15-05564-f003:**
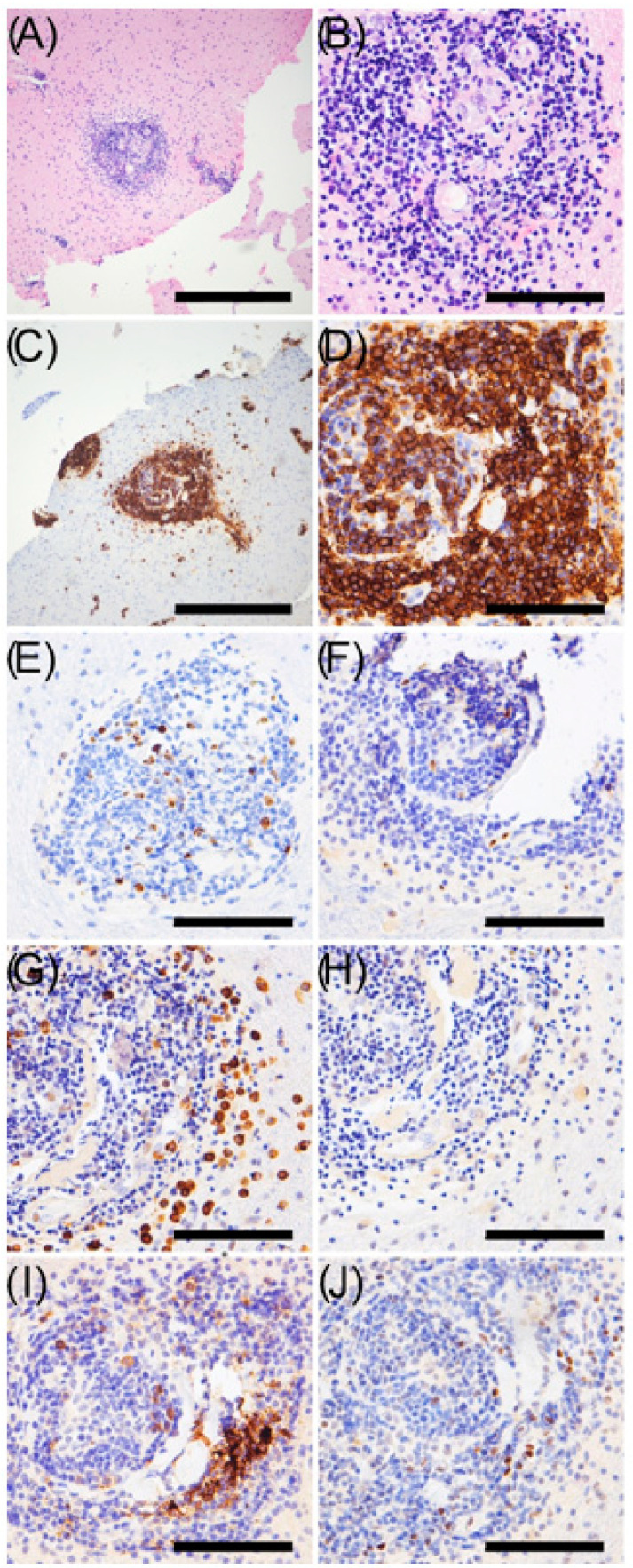
Patterns of cell distribution of a patient with plasmocytic differentiated marginal-zone lymphoma in the CNS. (**A**) Overview HE staining of perivascular lymphoma aggregates. (**B**) HE staining. (**C**) Overview of CD20 staining. (**D**) CD20 staining. (**E**) Ki-67 staining. (**F**) Cyclin D1 staining. (**G**) Kappa-ISH. (**H**) Lambda-ISH. (**I**) CD23 staining. (**J**) BCL6 staining. In the overview (**A**,**C**), the scale bar represents 400 µm. In the higher magnifications (**B**,**D**–**J**), the scale bar represents 8 µm. Courtesy of Prof. Klapper, University Hospital Schleswig-Holstein, Department of Pathology, Kiel, Germany.

**Figure 4 cancers-15-05564-f004:**
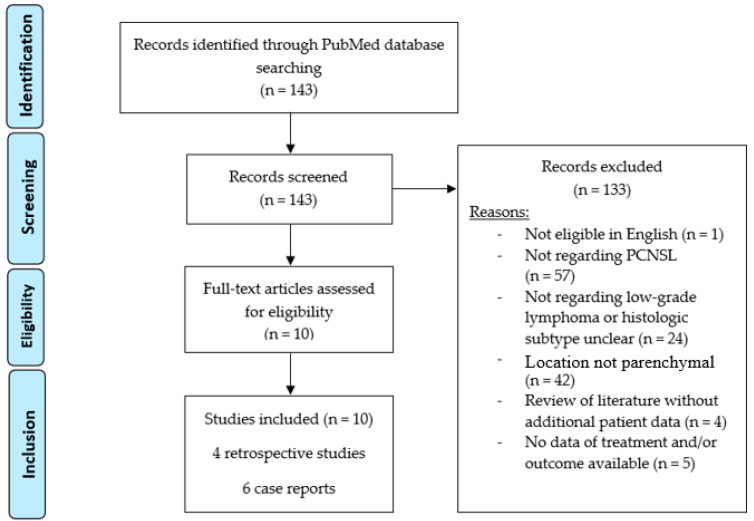
PRISMA diagram of the performed literature review in the NCBI’s PubMed.

**Table 1 cancers-15-05564-t001:** Demographics and data of the patients evaluated in this report. KPS: Karnofsky performance score, PR: partial remission, RTX: rituximab, SD: stable disease.

	Patient 1	Patient 2
Gender	Female	Female
Age of first diagnosis	53 years	57 years
Brain localization	Left frontal lobe	Left parietal lobe
RT concept	36 Gy ISRT	30.6 Gy ISRT
Number of fractions	20	17
Prior treatment	None	Four cycles of RTX
Treatment response (Follow-up duration)	SD (26+ months)	PR (14+ months)
KPS before treatment	80	70
KPS at last RT follow-up (months after RT)	80 (27)	80 (6)
Histological subtype	Marginal-zone lymphoma	Marginal-zone lymphoma
Clinical signs of neurological impairment after treatment	None	None
Radiological signs of treatment toxicity	None	None
EBV status	Negative	Negative
Immunohistochemistry	CD20+, CD45+, CD79a+, CD3−, CD5−, CD10−, CD23−, cyclinD1−, Lambda-LC-restriction, Ki-67 3% MYD88-wt, CXCR4-wt	CD20+, CD3−, CD5−, CD10−, MUM1+, BCL2+, BCL6−, CD30−, c-Myc− Ki-67 > 3%

## Data Availability

Please contact Niklas Benedikt Pepper regarding data availability.

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
