# Peer review of "Involved-Site Radiation Therapy Enables Effective Disease Control in Parenchymal Low-Grade Primary Cerebral Lymphoma"

_cancers, 2023, doi:10.3390/cancers15235564_

Round 1

Reviewer 1 Report

Comments and Suggestions for Authors

Manuscript of Pepper et al. presents a retrospective analysis of two patients treated with involved site radiotherapy and a review of the available literature on rare cases of low-grade parenchymal lymphoma.

Using an analysis of 2 cases of low-grade lymphoma, follicular lymphoma and marginal zone lymphoma, which are known to respond well to local radiotherapy [20,21], the authors conclude that aggressive chemotherapy may not be necessary to achieve a favorable outcome.

Main remarks

Because it has been known that LG-PCNSL are respond well to local radiotherapy (Elsayad at al Strahlenther Onkol 2020, 196, 117–125), novelty of this work is limited.

Although the authors argue that de-escalation of treatment in this form of PCNSL is necessary to avoid long-term toxicity, evidence supporting different regimens is limited due to the overall small cohort size.

Insufficient data for statistical analysis.

Minor remarks

The page numbering after page 8 is incorrect.

In general, the limited novelty, the small number of cases of LG-PCNSL and, as well as the lack of statistical data, raise doubts about the accessibility of the manuscript for publication in the journal, in the opinion of the reviewer.

Comments on the Quality of English Language

Minor editing of English language required

Author Response

Dear reviewer,

thank you very much for your feedback. We appreciate your support and expertise.

However, in response to your feedback we would like to emphasize the value of our work: while the effectiveness of RT is harnessed in several types of low-grade lymphoma (as pointed out in the work of Elsayad et al. you rightfully qouted), LG-PCNSL is a very rare entity with favourable outcome (as seen in the limited number of cases available in the literature), which is oftentimes treated with chemotherapy nevertheless. The small overall number of cases, limited to a handful of reports worldwide, casts the possibility of high-level evidence in the form of randomized controlled trials with statistical evaluations into doubt. Therefore, we try to enhance the available evidence by highlighting a favourable approach in the form of ISRT with discussion of the available literature. In the future, this article might not only offer better understanding of the possibilities of RT in this rare entity, but serve as a basis for building a data base to gather further data on this importatnt topic. Until then, statistical analysis in our work as in others published on that matter will not be possible, which is why it was excluded in our article. In our humble opinion, this does not negate the importance of the data.

Regarding your additional remarks, we fixed the numbering of pages.

Thank you again for your expertise.

Sincerely,

the authors

Reviewer 2 Report

Comments and Suggestions for Authors

The text relates with the favorable results of brain radiotherapy for low-grade lymphoma, in contrast with the adverse results with the high-grade lymphoma; I have only a minor suggestion, in keywords the term “irradiation” might the termed “radiotherapy”.

Comments on the Quality of English Language

No comments.

Author Response

Dear reviewer,

Thank you for your contribution, we gladly implement your suggestion regarding the re-phrasing of the term irradiation in the keywords.

Sincerely,

the authors

Reviewer 3 Report

Comments and Suggestions for Authors

The authors present an interesting study discussing low-grade primary parenchymal central nervous system lymphomas (LG-PCNSL), a rare entity with limited reported cases. The study concludes that parenchymal low-grade PCNSL demonstrates a good response to localized radiation treatment, leading to a favorable outcome while avoiding long-term treatment toxicity. The research study does not require additional data and it does an extensive and well-directed discussion of the literature.

Author Response

Dear reviewer,

thank you for your positive feedback on our article. We appreciate your support and expertise.

Sincerely,

the authors

Round 2

Reviewer 1 Report

Comments and Suggestions for Authors

The most common genetic alteration in PCNSL is the gain of chromosome 12 (12p12–14), which corresponds to the amplification of MDM2 to enhance p53 suppression.

Have you done genetic testing to check if there are genetic alteration in LG- PCNSL?

Due to the rarity of the LG-PCNSL form found in the world and the limited literature data, the article presented by the authors can be accepted in this form.